# Ontology Reasoning for Explanatory Feedback Generation to Teach how Algorithms Work [*]

Anton Anikin[1][0000−0003−0661−4284], Oleg Sychev[1][0000−0002−7296−2538], and Mikhail Denisov[1][0000−0002−1216−610X]

Volgograd State Technical University, Lenin Ave, 28, Volgograd, 400005, Russia
anton@anikin.name, oasychev@gmail.com

**Abstract.** Developing algorithms using control structures and understanding their building blocks are essential skills in mastering programming.In this work, we applied a formal model and reasoning rules for Jena reasoner to build execution trace for the given algorithm and find fault reasons if the student provides an incorrect answer. Using formal reasoner to check domain constraints allowed us to provide explanatory feedback for every type of error students can make.

**Keywords:** Ontology reasoning · Constraint-based tutoring systems · Program execution trace · Error detection.

## 1    Introduction

Ontology models and formal logic reasoning are used for knowledge representation and processing in different domains for a wide range of tasks. E.g., Ontology Driven Software Engineering (ODSE) approach [3] implies using ontology models for various aspects of software engineering: modeling different parts of software systems, products, modules, and algorithms. Most of these aspects are important in introductory programming courses as well.

One of the efficient approaches to introducing new learners to algorithms analysis and synthesis is the trace-based teaching approach that allows to decrease the dropout and grade failures by 25.49% and 8.51% respectively [1,2,4]. According to the structured programming approach, any algorithm can be represented as a tree of control structures. In the introductory programming teaching on the Problem Formulation step [2] the algorithmic reasoning skills ("a pool of abilities that are connected to constructing and understanding algorithms: to analyze given problems; to specify a problem precisely; to find basic actions that are adequate to the given problem; to construct a correct algorithm to a given problem using the basic actions" [5]) improvement is important. On the Solution Expression step, when the problem is formulated, and students should express a solution through programming structures, selecting the appropriate structures for solving the task is the main difficulty [6]. Finally, on the Solution Execution and Evaluation step, students should test and analyse the code to identify and

---

[*] The reported study was funded by RFBR, project number 20-07-00764.

correct problems, and code tracing activities is an appropriate task on this step [1].

So, automated algorithm-trace generation and analysis with explanatory feedback providing is an important task in introductory programming learning that can be solved using ontology domain models and formal logic reasoning. The reasoning rules allow to set the domain constraints and use it not only for the execution trace check for correctness but for the particular errors detection and corresponding explanation providing as well at the same time.

## 2 Intelligent Application to Teach Algorithms

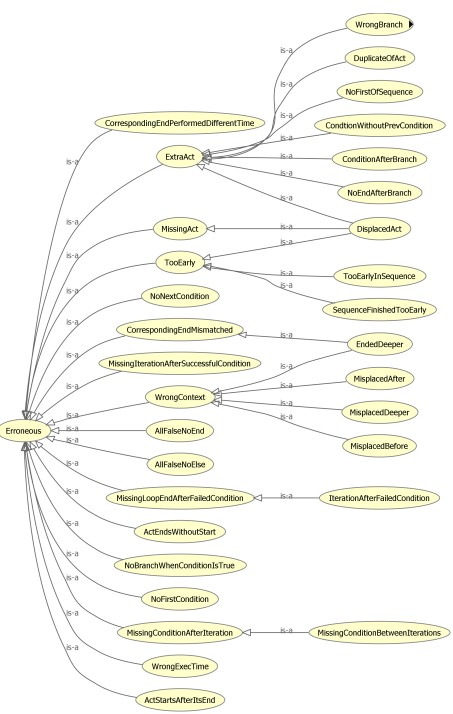

**Fig. 1.** Concepts for classifying errors

We developed an online tutoring tool `How It Works: Algorithms`[1] using ontology reasoning to grade students' answers and generate explanatory feedback about their errors. Its input consists of an algorithm, represented as a tree of basic control structures – sequences, alternatives, and loops (see Fig. 2) – and the values of control conditions. For grading purposes, the reasoner also receives the student-built trace as a sequence of control-statement execution acts. For complex control structures, the beginning and the end of their execution constitute separate act to represent the nesting of control statements in the trace.

To generate explanatory feedback, we classified all the possible errors in execution-trace building creating 33 concepts to represent them (Fig. 1). The reasoning engine determines the error class and the additional information about the individuals related to the error for feedback generation. We performed a study of software reasoners to find the best one for our domain, comparing Pellet, Apache Jena, Apache Jena SPARQL query processor, SWI-Prolog with semweb package, and ASP (Answer Set Programming) solvers Clingo and DLV. The results show that Apache Jena performs inference quicker than other reasoners on most of the domain-specific tasks.

In particular, Apache Jena infers the correct trace and student's errors 2.4–2.9 times quicker than SWRL Pellet reasoner. Jena rules also more expressive

---

[1] http://vds84.server-1.biz:3333/en/

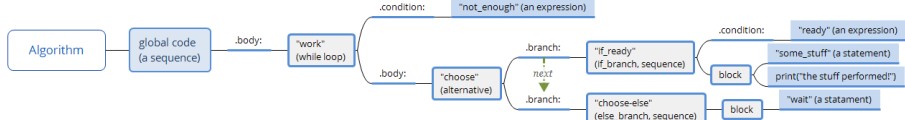

**Fig. 2.** Fragment of algorithm represented as an abstract syntax tree

than SWRL, having full CRUD operations support (e.g., creation of concepts and individuals), negation support, and relation retraction.

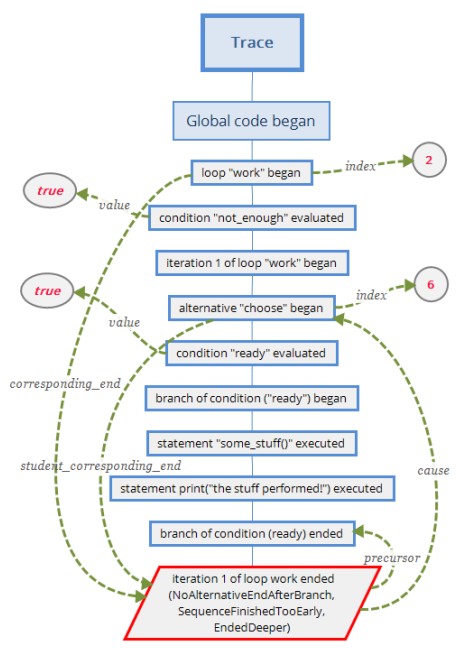

**Fig. 3.** Execution trace processing example

So, the developed ontology contains about 30 concepts for algorithm elements, 7 trace acts, 29 kinds of errors, and 27 explanations for correct acts; 30 roles, and about 100 positive, negative, and helper rules. Using these rules, the execution trace can be generated based given the algorithm, determine the domain constraints violated by the student, and provide all the necessary information about errors to generate explanatory feedback (Fig. 3).

The implemented software tool allows students to input the user execution trace for the teacher-defined algorithm (stored as an URL for easy access) by clicking buttons inside the algorithm and show detailed explanatory feedback for errors made using the ontology reasoning described above (Fig. 4).

## 3   Conclusion and Future Work

In this study, we present an ontology with a set of reasoning rules that is able to build execution traces for a given algorithm, find errors in students' traces, and provide the necessary information to generate explanatory feedback about the violated constraints representing subject domain laws. The approach was implemented in a software tool, using Apache Jena inference engine for ontology reasoning. The usage of forward chaining RETE algorithm and Jena rules and reasoning allowed us implementing domain-specific rules with adequate per-

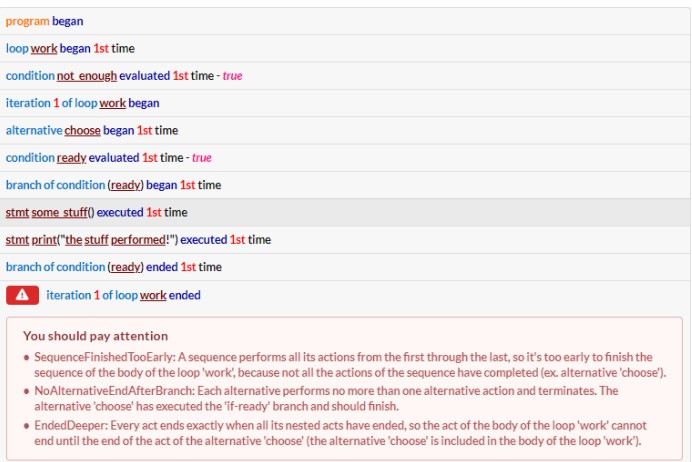

**Fig. 4.** Explanatory feedback provided by the develop tool for an error in the trace

formance to grade students' traces in real time step-by-step, showing feedback messages right after adding an erroneous line.

The software tool can be used as a basis for developing intelligent tutoring systems for improvement of algorithmic reasoning skills and developing understanding of program execution during introductory programming courses. The future work includes expanding the set of supported programming languages, supporting recursive functions in the algorithms, and developing a constraint-based intelligent tutoring system based on the proposed approach for complex exercises implementation by adding learner's model and intelligent exercise selection.

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
