# OpenReview forum: "Ontology Reasoning for Explanatory Feedback Generation to Teach how Algorithms Work"
_eswc-conferences.org/ESWC/2021/Conference/Poster_and_Demo_Track — Submitted to ESWC2021 P&D_

### Official Review · AnonReviewer1 · 2021-04-05
**the demonstrated tool should be more thoroughly documented**

**Rating:** 4
**Confidence:** 4

**Review:**

The demo should present a tool for learning programming whose novelty is its use of an ontology for classifying errors. The application presented in the paper is relevant but too few details are given either on the added value of Semantic Web technologies or on the effectiveness and reusability of the tool itself. I therefore suggest to further document your work and resubmit at a later conference.

Remarks on the content:
 - what is the level of education targeted by the tool? Ref [2] mentions "higher education", ref [1] mentions "early programming courses". Blockly, the UI library used in the tool, is also meant for lower education levels, as far as I know.
 - the provided citations do not clearly point to what is part of the state-of-the-art and what is novel. For instance, where does the three-steps learning process come from (problem formulation, solution expression, solution execution and evaluation)? Is it proper to the demonstrated tool? Moreover, if trace-based teaching improves the students' understanding, aren't there tools that already exist in the area?
 - the ontology doesn't seem to be publicly available. In particular, no example of rule is given in the paper, which makes it hard to evaluate what benefit these rules have on providing feedback to students.
 - could there be quantitative results on the different rule engines mentioned in the paper? What are e.g. the absolute execution times?
 - the paper includes no evaluation of the effectiveness of the reasoning task. That aspect could be part of a broader line of work to be evaluated later but, then, it should be more clearly emphasized what the demo brings to the state-of-the-art.

On the writing and the flow of the paper:
 - figure 2 is introduced before figure 1. Could they be switched?
 - the conclusion introduces new elements ("real-time" reasoning for direct feedback?). These elements could rather be part of section 2.
 - the paper often uses the form "something doing" instead of "doing something" (e.g.: "programming learning", "execution-trace building").

**Anonymity:**

Yes, I would like my review to remain anonymous.

---

### Official Review · ~Julien_Corman2 · 2021-04-13
**The underlying work seems very interesting, but the article fails to explain (even informally) what the system does.**

**Rating:** 4
**Confidence:** 3

**Review:**

The article describes an apparently complex rule-based system, used to teach algorithmics.
However, I feel like it fails to explain (even informally) what the system does.

My guess is the following: students are given an algorithm, and asked to write an execution trace for this algorithm.
Then a rule engine is used to check whether a trace written by a student is valid for this algorithm.
If not, the rule engine is also able to classify (one or several?) error(s?) made by the student.
Alternatively, the (same?) rule engine may be used to infer the correct execution trace for a given algorithm and input.

However, this is nowhere said explicitly, so these are blind guesses, from clues gathered throughout the paper.

I understand that space limitation do not allow for a formal description.
But space could be gained by dropping the didactic arguments in Section 1, which are only meant to legitimate the approach (dropout and grade failures, or the "Problem Formulation" and "Solution Expression" steps, which are irrelevant for this work).

Figure 3 also raises a lot more more questions than it helps (see the list of questions below).
Maybe providing instead a small selection of rules used by the engine would give a better intuition of what the system actually does.

A slightly more formal characterization of the rule engine would also be welcome (reasoning tasks, semantics, expressivity of the rules), in particular for a Semantic Web conference.
I guess these rules are non-recursive, because the authors considered using SPARQL query evaluation as an alternative.
But what does "negative rules" (page 3) exactly mean?
And under which semantics are they interpreted?
On a similar note, the article mentions (twice) "domain constraints" violations, but it is unclear how these are identified.

Overall, a lot of confusion could be avoided with a more structured presentation.
For instance, the inference engine is described in Section 2, whereas the RETE algorithm is only mentioned in conclusion.
Similarly, Section 2 describes part of the ontology, then the inference engine, and then the rest of the ontology.

A possible structure could be:
1) components of the system, with a clear list of tasks (input/output) executed by each component,
2) format of the input algorithms (trees of basic control structures), format of the traces (input traces and generated traces),
3) ontology,
4) rule engine: reasoning tasks, algorithms, expressivity, and maybe a classification of rules by usage (trace verification, trace generation, classification of errors, etc.), with small samples
5) user interface

To summarize, the underlying work seems very interesting, but the paper does not do justice to it in my opinion.
I suggest either an important refactoring of the paper, or maybe opting for a more appropriate publication format (not sure a demo track is the right venue).
There seems to be enough underlying work here for a journal publication, where the system could be described in a more detailed and structured fashion.


## Questions

- Page 2: "set the domain constraints and use it".
What does "it" refer to here?

- Page 2: "and algorithms, represented as [..] and the values of control conditions".
I guess "value" means true/false.
But an algorithm should not have values for its control conditions (a trace on the other hand should).
Or does "value" mean something else here?

- Page 2: "For grading purposes, the reasoner also receives the student-built trace".
"For grading purposes" suggests that the reasoner could also be used without such a trace as input, if the purpose is not to grade a student.
But for what other tasks exactly?

- Page 2: "constitute separate act".
What does "act" mean?

- Page 2: "Apache Jena infers the correct trace".
I guess this means the correct trace for a given input.
But why would one need a rule engine to produce such a trace?
Isn't it sufficient to execute the program with this input?

- Figure 3.
I am really confused by this figure.
This looks like a (form of) execution trace, i.e. a sequence of executed instructions.
However, the title suggests that this is not a trace, but the "processing" of a trace (I do not understand what this means).
The top element of the diagram (the square box "Trace") is also confusing.
It suggests that the sequence of events starts with the trace itself (which does not make sense to me).
I also wonder whether the green arrows are exhaustive, or just a sample of relations that may hold between events.
For instance, why do two instructions only have an index?
And what is the difference between "corresponding_end" and "student_corresponding_end"?
I would recommend either dropping this figure, or providing an explanation of what it is meant to illustrate.

- Page 3: "27 explanations for correct acts".
What is a correct act, and why does it need an explanation?

- Page 3: "positive, negative, and helper rules".
Are "helper rules" neither positive nor negative?
And what exactly is a "negative rule" (negation in the body, OWL disjointness axiom, ..)?

- Figure 4: "provided by the develop tool".
What is "the develop tool" (this is the only mention of it in the paper)?

- Conclusion: "expanding the set of supported programming languages".
This is the first mention of a "programming language".
Which ones are already supported (the web interface lists C++, Java and Python)?
Or are maybe trees of basic control structure considered as a programming language (this is fine, but it should be made explicit)?


## Suggestions

- Abstract: "for every type of error students can make".
This seems a little bit ambitious.
Maybe "for most types of errors"?

- Page 2: "stored as an URL for easy access".
Not sure this is a useful piece of information.
Also wondering what "stored" means.
Is the algorithm actually serialized as an URL?


## Typos:

- Abstract:
"Using formal reasoner" -> "Using formal reasoning" or "Using a reasoner"

Page 1:
"Ontology Driven Software Engineering (ODSE) approach implies" -> "the Ontology Driven Software Engineering (ODSE) approach implies"

- Page 2:
"code tracing activities is an appropriate task" -> "code tracing is an appropriate task" or "code tracing activities are appropriate"

- Page 2:
"algorithm-trace" -> "algorithm trace"

- Page 2:
"constitute separate act" -> "constitute separate acts"

- Page 2:
"with explanatory feedback providing" -> "with explanatory feedback"

- Page 2:
"Jena rules also more expressive" -> "Jena rules are also more expressive"

- Page 3:
"generated based given the algorithm" -> "generated based on the algorithm"

- Page 3:
"allowed us implementing" -> "allowed us to implement"

**Anonymity:**

No, I would like my review to be deanonymized.

---

### Official Review · AnonReviewer4 · 2021-04-14
**Interesting idea but lacking in description and execution**

**Rating:** 3
**Confidence:** 3

**Review:**

This demo intends to use semantic web technologies to describe program structure and execution, supporting algorithm design/analysis in computer science education.

From a 30'000 feet perspective the idea is very good and well worth pursuing. Unfortunately however, I find that the submission falls apart in the details.

1. The solution, the ontology, and the implementation is described in a much too limited fashion. This is probably a natural consequence of the page limit. Some central concepts are glossed over, that could be covered in substantially more depth, had the pages been available, included but not limited to:

    a) the structure of the primitives used as input (the tree of control structures is simple enough to grasp, but what is meant by "trace as a sequence of control-statement execution acts", and where do I see this in the web GUI provided?)
    b) the reasoner performance evaluation described in section 2
    c) the structure of the ontology and the rules employed to execute the system

2. There are some issues with readability; e.g., the last two sentences of the introduction section (the last of which I cannot, after multiple attempts, parse the meaning of), the figures are due to scale unreadable in print. Also, I would recommend the authors utilize standard nomenclature for their control loop structures, i.e., "Alternative" -> "Selection", "Loop" -> "Iteration"

3. Attempting to use the web based tool to model a simple bubble sort, I note that

    d) There seem to be no GUI components supporting the setting of variables. Is this supposed to be written in the name field of a custom Action?
    e) In spite of the above, the tool generates a written execution description when executing the algorithm I input, even tough I have given it no input on which to operate. This does not seem conducive to good teaching -- it should probably throw an error of some sort?
    f) The language selection GUI appears to have no impact what-so-ever on the resulting generated description. Given that the tool focuses on control structures rather than language-specific syntax, I don't really understand why this GUI is there at all?

**Anonymity:**

Yes, I would like my review to remain anonymous.

---

### Official Review · ~Edelweis_Rohrer1 · 2021-04-14
**The presented approach appears as useful tool to students and beginner developers in their first steps. However, the model behind the tool is not clearly introduced.**

**Rating:** 6
**Confidence:** 4

**Review:**

Summary

With the motivation of helping students to develop algorithms and provide adequate feeback when errors occur, this paper presents an ontology, reasoning rules and a process that are implemented in an online tutoring tool for beginner developers. The implemented tool receives an algorithm and input values, and build execution traces, generating explanatory feedback about errors.

Evaluation

Section 2 describes at a higher level the process followed to generate detailed feedback for the developer. However, there are some points in the description of Section 2 that are not clear:
-	A taxonomy of possible errors is mentioned and also an ontology that assist the process to generate the feedback. Are they different ontologies?
-	If so, what is exactly the role that plays each element of the latter ontology?
-	The paper does not give any idea about how this ontology is conceptualized, and at least an example of the associated
reasoning rules, i.e. how starting from the tree that represents the algorithm, the information to generate feedback is inferred.

Final comments

The presented approach appears as a useful tool to students and beginner developers in their first steps. However, the approach is not clearly introduced; in particular, the ontology and rules, that are the core part of the system, are not properly described.


**Anonymity:**

No, I would like my review to be deanonymized.

---

### Decision · Program_Chairs · 2021-04-19

Reject